# Optimising global landscape evolution models with $^{10}$Be

Gregory A. Ruetenik[1], John D. Jansen[1], Pedro Val[2], Lotta Ylä-Mella[1,3]

[1] GFÚ Institute of Geophysics, Czech Academy of Sciences, Prague, Czech Republic

[2] School of Earth and Environmental Sciences, Queens College, City University of New York, New York, USA

[3] Department of Physical Geography and Geoecology, Charles University, Prague, Czech Republic

*Correspondence to*: Gregory A. Ruetenik (ruetenik@ig.cas.cz)

**Abstract**

By simulating erosion and deposition, landscape evolution models (LEMs) offer powerful insights to Earth surface processes and dynamics. Stream-power based LEMs are often constructed from parameters describing drainage area ($m$), slope ($n$), substrate erodibility ($K$), hillslope diffusion ($D$), and a critical drainage area ($A_c$) that signifies the downslope transition from hillslope diffusion to

advective fluvial processes. In spite of the widespread success of such models, the parameter values are highly uncertain mainly because the advection and diffusion equations amalgamate physical processes and material properties that span widely differing spatial and temporal scales. Here, we use a global catalogue of catchment-averaged cosmogenic $^{10}$Be-derived denudation rates with the aim to optimise a set of LEMs via a Monte Carlo based parameter search. We consider three model scenarios:

advection-only, diffusion-only, and an advection-diffusion hybrid. In each case, we search for a parameter set that best approximates denudation rates at the global scale, and we directly compare denudation rates from the modelled scenarios with those derived from $^{10}$Be data. We find that optimised ranges can be defined for many LEM parameters at the global scale. In the absence of diffusion, $n \sim 1.3$, and with increasing diffusivity the optimal $n$ increases linearly to a global maximum

of $n \sim 2.3$. Meanwhile, we find that the diffusion-only model yields a slightly better correlation than the advection-only model and is optimised when the concavity parameter is raised to a power of 2.

With these examples, we suggest that our approach provides baseline parameter estimates for large-scale studies spanning long timescales and diverse landscape properties. Moreover, our direct comparison of model-predicted versus observed denudation rates is preferable to methods that rely

upon catchment-scale averaging or amalgamation of topographic metrics. We also seek to optimise $K$ and $D$ parameters in LEMs with respect to precipitation and substrate lithology. Despite the potential bias due to factors such as lithology, these optimised models allow us to effectively control for topography and target specifically the relationship between denudation and precipitation. All models suggest a general increase in exponents with precipitation in line with previous studies. When isolating

$K$ under globally optimised models, we observe a positive correlation between $K$ or $D$ and precipitation > 1500 mm yr$^{-1}$, plus a local maximum at $\sim$ 300 mm yr$^{-1}$, which is compatible with the long-standing hypothesis that semi-arid environments are among the fastest-eroding landscapes.

## 1 Introduction

To appreciate short-term changes in Earth surface processes, such as those induced by humans (Brown, 1981; Hooke, 2000), it is first necessary to understand long-term rates of denudation and deposition. Recognizing this, some recent studies (e.g., Simoes et al., 2010) derive erosion-transport rules from topography with an aim to predict macro-scale patterns of denudation and sediment flux. At more restricted spatial scales, denudation rates based on cosmogenic nuclides (e.g., $^{10}$Be) show a

modest exponential correlation with catchment-averaged slope, as does normalised steepness in stream profiles (Portenga and Bierman, 2011; Harel et al., 2016). Nevertheless, it is widely observed that steepness and stream power parameters are subject to considerable variation wherever climate and/or lithology differ (Harel et al., 2016; Gailleton et al., 2021; Marder and Gallen, 2022), and that the parameters also covary. A robust analysis must accommodate such interactions.

Earth's surface undergoes continuous modification through uplift and denudation over timescales too long to observe directly, hence landscape evolution models (LEMs) are crucial tools for building knowledge. Note that we use the term 'denudation' to denote the mass loss leading to lowering of Earth's surface. LEMs are often employed over expansive scales of space and time in

order to study topographic response to changes in tectonics (e.g., Kooi and Beaumont, 1996; Garcia-

Castellanos et al., 2003), climate (e.g., Temme et al., 2009, Adams et al., 2020), and sea level (e.g.,

Pico et al., 2019; Ruetenik et al., 2019). And yet, large spatial and temporal scales require

generalisation of model parameters that accounts reliably for processes of hillslope diffusion and

advective fluvial erosion. Using LEMs to estimate denudation rates delivers the key advantage of

bridging scales and defining an empirically derived mechanism at the local (grid cell) scale. This

demands that denudation rates are integrated over scales matching the topographic changes they

describe. At the local to regional scale, recent studies have focused on constraining LEM parameters

via inversions that optimise rates of denudation, deposition, and topographic observations (e.g., Miller

et al., 2013; Croissant and Braun., 2014; Pedersen et al., 2018; Barnhart et al., 2020). However,

implementing many of these approaches at a global scale is challenging in terms of both

computational cost and because it often requires a compilation of a large set of observables (such as

knickpoints, depositional patterns, and denudation rates). In the absence of computational power that

can accurately simulate stratigraphy at the global scale, and without constraints on global palaeo-

topography, we settle for optimising LEM parameters with respect to catchment-averaged denudation

rates estimated with cosmogenic $^{10}$Be. While LEM parameters can be estimated via direct topographic

analysis (e.g., Wobus et al., 2006, Clubb et al., 2014, Mudd et al., 2018), this approach can lead to

bias, as we discuss below.

Here, we determine LEM parameter values that minimise the variance among $^{10}$Be-derived

apparent denudation rates in the *OCTOPUS* v.2 global catalogue (Codilean et al., 2022), and we

analyse the capacity of LEMs to predict denudation rate given those optimised parameters. Our LEM

employs the common stream-power-plus-diffusion formulation, which is subject to important

limitations, such as the neglect of fluvial deposition and mass wasting processes (e.g., Whipple and

Tucker, 1999). The trade-offs involved in this simplified approach, we believe, are justified by the

record of success with simulating landscape processes at large scales and across a wide range of

lithologies, drainage areas, and steepness (e.g., Gallen et al., 2013; Miller et al. 2013; Fox et al., 2014).


## 1.1 Catchment-averaged denudation rates from cosmogenic [10]Be

Rates of catchment-scale denudation can be estimated by measuring the abundances of cosmogenic radionuclides, such as [10]Be, in quartz-bearing river sand (Granger et al., 1996; von Blanckenburg, 2005). Such nuclides accumulate within minerals exposed to secondary cosmic rays in the upper few metres of the bedrock subsurface and are lost via erosion and radioactive decay (Lal, 1991). The attenuation of cosmic rays with depth causes the nuclide production rate to decrease exponentially (at 2 m depth the [10]Be production rate is < 5 % that at the surface); hence, nuclide abundances measured in sediment are an inverse function of denudation rate.

The spatial variations observed in denudation rates across a range of climates and lithologies (Portenga and Bierman, 2011; Starke et al., 2020) suggest that the erosional processes driving the evolution of landscapes also vary. This has important implications for the interpretation of [10]Be-derived denudation rates and how we parameterise LEMs. Estimating catchment-averaged denudation rates from nuclide abundances in river sand depends on a multitude of assumptions (von Blanckenburg, 2005; Mudd, 2016). For example, we assume sediments were produced via long-term, steady bedrock erosion distributed uniformly across the catchment, and that sediments have experienced continuous exposure to cosmic rays at/near the surface. In detail, long-term steady erosion refers to at least one attenuation length (~ 0.6 m) of surface lowering integrated over a $10^3$–$10^5$ yr timescale. Abrupt bedrock erosion events, for instance, caused by bedrock landsliding or glacial quarrying may bias denudation rate estimates. Similarly, long intervals of ice cover or intermittent deep sediment burial contradict the requirement for continuous cosmic-ray exposure. Other sources of potential discord relate to lithology, catchment size, and hypsometry, which are known to affect sediment transport dynamics and grain-size yields (Carretier et al., 2015; Riebe et al., 2015; Lukens et al., 2016; Zavala et al., 2020). The sources of deviation noted above are collectively responsible for the considerable variability observed in large compilations of [10]Be-derived denudation rates (e.g., Portenga and Bierman, 2011; Harel et al., 2016). Catchment-wide denudation rates are commonly determined and published for settings that do not comply strictly with the method's premises; these

estimates are best referred to as *apparent* denudation rates (Mudd, 2016). Nevertheless, [10]Be-derived data currently offer the most widely distributed insight to long-term denudation on a global scale.

## 2 Methods

### 2.1 Stream power and hillslope diffusion

Stream power is represented by a non-linear advection equation derived from observations of river morphology and generalised relationships for bed shear stress (e.g., Howard et al., 1994; Whipple and Tucker, 1999; Lague, 2014). It affords a description of channel incision as a function of upstream drainage area (*A*) and local slope (*S*) for the portion of the catchment (above the critical drainage area, $A_c$) where fluvial advection dominates over hillslope diffusion and debris flow processes (e.g., Lague and Davy, 2003; Fontana et al., 2003; Whipple and Tucker, 1999). The stream power equation takes the form:

$$E_{predicted,advective} = K A^m S^n , \qquad (1)$$

where *K* is the advection coefficient or erosional efficiency that is (in our formulation) uniform across a catchment, and *m* and *n* determine the relative dependence of incision on drainage area and slope. The ratio *m/n* defines how elevation scales with drainage area at steady state relative to the longitudinal channel profile and typically varies from 0.3 to 0.6 (Wobus et al., 2006; Whipple and Tucker, 1999); *n* determines the erosional nonlinearity, which is thought to relate to regional discharge variability, as well processes governing incision thresholds, and typically ranges from 1 to 4 (e.g., Lague, 2014). A global compilation of stream power parameters constrained by topographic metrics (Harel et al., 2016) reports an optimised *n* ~ 2.6, albeit with considerable scatter (also observed by Gailleton et al., 2021). The value of *n* may also vary depending on the location in the channel network —the steepest and fastest-eroding locales such as knickpoints can have values closer to unity (Lague, 2014). In general, higher *n* results in larger erosional flux from steep terrain, while higher *m* results in

larger erosional flux from big rivers. However, if $m$ and $n$ both increase (keeping their ratio, and $K$ constant), a larger fraction of erosional flux will be sourced from steeper main-stem channels.

The amalgamated outcomes of hillslope transport processes, such as rainsplash, soil creep, and bioturbation, are primarily diffusive. Hence, to simulate hillslope processes, we include a diffusion equation:

$$E_{predicted,diffusive} = D \left( \frac{d^2 z}{dx^2} + \frac{d^2 z}{dy^2} \right)^p, \qquad (2)$$


where $D$ is diffusivity, which is reported to range from $\sim 4.4 \times 10^{-4}$ to $3.6 \times 10^{-2}$ for linear diffusion (e.g., Fernandes and Dietrich, 1997). We use an exponentiated form of the diffusion equation in which concavity is raised to an exponent, $p$, in order to harmonise with Gabet et al. (2021), which posits that denudation rate scales approximately with hillslope concavity squared. For linear diffusion to satisfy

mass conservation, the deposited and eroded sediment should balance. However, in the exponentiated formulation, negative concavities raised to a power of $p$ can produce non-real components. Thus, we calculate an average catchment-wide denudation rate in which we ignore deposited sediment (areas of negative concavity) and take an average based on eroded sediment only.

        For our joint advection-diffusion model, we follow the common approach of combining stream

power with linear diffusion (i.e., $p = 1$ in Eq. 2). When Equations (1) and (2) are combined, $D$ and $K$ covary, and it has been noted that a higher $D/K$ ratio results in a more linear scaling between erosion rate and catchment-averaged slope (Forte et al., 2016). Hence, we divide (Eq. 3) by $K$, which allows the $D/K$ ratio to be optimised with respect to predicted erosion rate:

$$\frac{E_{predicted}}{K} = A^m S^n + \frac{D}{K} \left( \frac{d^2 z}{dx^2} + \frac{d^2 z}{dy^2} \right), \qquad (3)$$

With our advection-diffusion model formulation (Eq. *3*), we set out to solve simultaneously for globally optimised values of $D/K$, $n$ and $A_c$. $D/K$ and $n$ determine the relative importance of advective versus diffusive processes in driving erosion; lower $D/K$ and higher $n$ (which implies higher $m$ given uniform $m/n$) will result in the dominance of advection, whereas higher $D/K$ and lower $n$ will promote diffusion. While varying $m$ and $n$, we maintain their ratio constant at 0.45, a widely applied average channel concavity (e.g., Wobus et al., 2006; Harel et al., 2016), and in line with the global average of 0.42 reported by Gailleton et al. (2021) (despite considerable variability).

Based on previous modelling (e.g., Roering and Kirchner., 2007), one might expect advection-dominant landscapes to be rougher in outline relative to the smoothing effects of diffusion. However, Theodoratos et al. (2018) show that multiple sets of parameters can give rise to equifinality. In our modelling framework, the $D/K$ ratio covaries with $n$ to determine the ratio of hillslope denudation and total (fluvial + hillslope) denudation—denoted here as $E_{predicted,diffusive}/E_{total}$, where $E_{total}$ is the sum of $E_{predicted,diffusive}$ and $E_{predicted,advective}$. The ratio $E_{predicted,diffusive}/E_{total}$ is therefore a function of both $n$ and $D/K$. In principle, this metric is inversely proportional to an effective Peclet number for net denudation (Perron et al., 2008). For values of $E_{predicted,diffusive}/E_{total}$ close to unity, diffusive processes will dominate, while values closer to zero represent advection dominance (Fig. 1).

## 2.2 [10]Be-derived apparent denudation rates

We conduct modelling experiments that employ randomly selected sets of LEM parameter values, and then compare our model outputs with the global catalogue of [10]Be-derived catchment-averaged denudation rates, *OCTOPUS* v.2 (Codilean et al., 2022). In addition to apparent denudation rates (N = 4631), *OCTOPUS* includes topographic data and catchment outlines. Using the catchment boundaries in *OCTOPUS*, we clipped rasters from the *Hydrosheds* Shuttle Radar Topography Mission (SRTM) dataset, a global digital elevation model (DEM) with 3-arc-second resolution (Lehner et al., 2008), plus the National Elevation Dataset (Gesch et al., 2002) for catchments in Alaska north of 60 degrees. Local pits (i.e., lakes) within the catchments were filled using the priority-flood method of Barnes et al. (2014). In building the network of local upstream drainage areas for each cell, runoff is assumed to

flow down the steepest descent in accordance with the *D8* flow routing algorithm. Slopes for every cell are computed along this steepest-descent flow path.

To determine the effect of DEM resolution, we test our models on 1-arc-second Copernicus DEM data, although for reasons of computational capacity we restrict our analyses to catchments with < 1.3 million grid cells (N = 3414), and for the diffusion-only and advection-only scenarios. We find that DEM resolution is not decisively important to our results (Supplementary figs. S1, S2).

About 24 % of the *OCTOPUS* dataset is not exploitable for our purposes: 68 catchments are too small to be processed by the LEM (< 3 DEM cells in any dimension), and we exclude 33 of the very largest catchments due to the extreme computational cost. Multiple denudation rate measurements included in *OCTOPUS* refer to samples from different locations within the same larger catchment. For such cases, a separate catchment is defined only where the drainage area differs by > 5 %; otherwise, we amalgamate the data and derive a single average denudation rate. In total, 3640 catchment-wide apparent denudation rates ($E_{apparent}$) are used in our modelling experiments, ranging from 0.028 km$^2$ to 430,000 km$^2$ (11 to 53 million grid cells). We do not separate $^{10}$Be measurements conducted on different grain-size fractions.

Processing DEMs begins with computing the drainage network. To minimize the effects of disequilibrium perturbations (e.g., Schwanghart and Scherler, 2017), we smooth river profiles using a ~1 km averaging window. Un-smoothed profiles are also used, for comparison, and yield similar results, where the largest discrepancies lie in the advection-diffusion model. After catchment slopes and drainage areas are computed, the diffusion and stream power equations can be solved. The LEM is run for exactly one time-step on DEMs representing each of the 3640 catchments; the sediment flux to the catchment outlet is then averaged over the total drainage area to yield a LEM-predicted denudation rate ($E_{predicted}$). Because different values of input LEM parameters typically yield different denudation rates (with the exception of highly diffusive models, as described below), such models are then optimised by comparison with our catalogue of $E_{apparent}$ data.

## 2.3 Monte Carlo simulations

We use a brute-force Monte Carlo approach to investigate the parameter space by running randomly selected sets of parameters and testing the fit of modelled versus observed ([10]Be-derived) denudation rates. We adopt the philosophy of equifinality (e.g., Beven and Binley, 1992) to evaluate the model parameters applied in our LEMs; implicit in these assumptions is that multiple sets of parameters may give rise to a similar, or equifinal, result (e.g., Csilléry et al., 2010). Hence, we report both the range of optimal parameters in addition to the best-fit model parameters.

In our framework, no more than three parameters are modified and compared at any one time (Table 1). This is possible thanks to several simplifying steps (detailed below) that require fewer modelling runs (e.g., Theodoratos et al., 2018). The performance of the model with a given set of parameters is evaluated based on the mismatch between $E^*_{predicted}$ and $E_{apparent}$ with respect to the likelihood function (Beven and Binley, 2014). Modelled and observed rates are compared directly, no regression is involved. In so doing, one or more local maxima representing an optimised parameter set may be identified in the space defined by parameter values versus the likelihood function. A range is then defined within 1 % of the peak (for example, if the best-fit model has a score of 0.500, we report the range of parameters from models with scores > 0.495.

For each randomly selected set of parameter values (Table 1), the LEM computes a single time step, and the erosion in each grid cell is integrated. $E^*_{predicted}$ is then scaled by employing a log-transformation on all modelled catchments:

$$\log\left(K^*\right) = \frac{1}{N} \sum \left( \log\left(\frac{E_{predicted}}{K}\right) - \log\left(E_{apparent}\right) \right), \qquad (4)$$

$$\log\left(E^*_{predicted}\right) = \log\left(\frac{E_{predicted}}{K}\right) + \log\left(K^*\right), \qquad (5)$$

where $N$ is the number of observations and $K^*$ is $K$ prior to log-transformation (see Eq. *7* below). The

performance of $log(E^*_{predicted})$ against $log(E_{apparent})$ is then calculated for each parameter set. This setup

offers the advantage of limiting the number of parameters varied; it is not our aim to determine the

absolute values of coefficients because these covary; instead, we focus on the ratio $D/K$ (see Section

2.1).

       After optimal values of $A_c$, $D/K$ and/or $n$ are found, the associated value of $K$ can be corrected

for log-transformation using an unbiased estimator (after Ferguson, 1986):

$$K = K^* e^{\frac{s^2}{2}}, \qquad (6)$$

where $s^2$ is an estimate of the variance:

$$s^2 = \frac{1}{N-1} \sum \left( \log\left( E_{apparent} \right) - \log\left( E^*_{predicted} \right) \right)^2, \qquad (7)$$

       An equivalent form of Eqs. *4–7* is used for the diffusion-only model, simply by replacing $K$

with $D$. Although we believe this log transformation is justified, we also provide coefficients without

Eqs. 6–7 (Supplementary fig. S7c).

Given that our denudation rate data span several orders of magnitude, we compare $log(E_{apparent})$

and $log(E^*_{predicted})$ using the Nash-Sutcliffe coefficient of efficiency (NSE):

$$NSE = 1 - \frac{\sum \left( E_{apparent} - E^*_{predicted} \right)^2}{\sum \left( E_{apparent} - mean\left( E_{apparent} \right) \right)^2} \qquad (8)$$

       Optimised values are defined as the maximum NSE value only where they are surrounded by

local maxima in the likelihood, which is the case in all experiments below. We also tested a version of

NSE that incorporates measurement uncertainty, based on Harmel and Smith (2007), but it had little

effect on the optimised parameters (Supplementary fig. S3, S4). Additionally, we calculate the Mean

Absolute Error (MAE) to gauge the sensitivity to the likelihood function (Supplementary fig. S5, S6).

### 2.4 The influence of lithology and precipitation on denudation

We subdivide the *OCTOPUS* catchments according to (1) areally-dominant lithology based on the GLiM global geologic map (Hartmann and Moosdorf., 2012), which gives a vectorized description of lithology compiled from a number of regional high-resolution geologic maps at a target resolution of 1:1,000,000; and (2) spatially-averaged mean annual precipitation (MAP) using the CHELSA dataset (Karger et al., 2016).

Precipitation differences between lithologic subgroups can be significant; for instance, averages range from 851 to 2077 mm yr$^{-1}$ for unconsolidated sediment and metamorphic rocks, respectively. To address lithological variations in the presence of climatic differences between lithologic subgroups in the advection-only model, we attempt to isolate substrate effects with the form of the stream power equation given by Kooi and Beaumont (1996), which explicitly includes precipitation variations that are normally folded into *K* under the assumption that precipitation (*P*) scales linearly with discharge:

$$E_{predicted,advective} = K_{lith}(PA)^m S^n, \qquad (9)$$

Although many factors influence *K* besides precipitation, we use $K_{lith}$ in this case to denote the variable we are attempting to isolate. We do not attempt to correct for variable precipitation in calculating *D*; for instance, by devising an equivalent $D_{lith}$ from Eq. 9, and it is only applied when calculating *K*.

### 3 Results

### 3.1 Advection-only model

We apply a stream power-based advection-only model (eq. 1) (excluding hillslope diffusion), with two free parameters: a slope exponent (*n*) and critical drainage area ($A_c$). Variations in *m* are fixed to *n* such that concavity is held constant at $m/n = 0.45$. We report the optimised values in terms of maximum value and an optimised range of values ($Q_{0.01}$) that are within 1 % of the maximum. The advection-only model (Fig. 2) is globally optimised at $n \sim 1.28$ ($Q_{0.01} = 1.23–1.43$; Fig. 2a) and at $A_c \sim$ 0.05 km$^2$ ($Q_{0.01} = 0.03–0.07$ km$^2$; Fig. 2b). We note that *n* changes by ~6 % ($n \sim 1.36$) using the higher-

resolution 1-arc-sec DEMs (Supplementary fig. S1). The slight differences in $n$ are likely the result of

the sensitivity to higher catchment-averaged slopes, which naturally arises from higher resolution

topographic data (Supplementary Table S1).

### 3.2 Diffusion-only model

The diffusion-only model (Fig. 3) is globally optimised with the hillslope diffusion exponent, $p \sim 2.0$

(NSE = 0.51, Fig. 3b), and with negligible dependence on DEM-resolution; $p \sim 2.0$ for the 1-arc-sec

models (Supplementary fig. S2). We also find that the 1-arc-sec Copernicus DEM has a lower overall

performance than the 3-arc-sec SRTM DEM (see discussion in Supplementary fig. S2).

### 3.3 Advection-diffusion model


While the optimisation of our diffusion-only model with $p = 2$ (Fig. 3) is an intriguing result that

invites further investigation (cf. Gabet et al., 2021), we retain linear diffusion ($p = 1$) in our advection-

diffusion experiment because for $p \neq 1$ the model is (1) numerically unstable when implemented in

LEMs, and (2) it fails to accommodate hillslope deposition and hence does not conserve mass.

The advection-diffusion model (Fig. 4) is globally optimised at $n \sim 2.3$ ($Q_{0.01}$ = 1.59–2.70; Fig.

4c), and $D/K \sim 1.79 \times 10^6 \ m^{0.9n+1}$ ($Q_{0.01} = 2.39 \times 10^4$ to $3.44 \times 10^7$; Fig. 4d). For $n$ and $D/K$, the $Q_{0.01}$

ranges are quite broad in part because both parameters are co-dependent (Fig. 4a). Optimum $A_c \sim 0.03$

$km^2$ ($Q_{0.01}$ = 0.01–0.17; Fig 4e) is similar to that from the advection-only models. The models are more

diffusive when $n$ is low and $D/K$ is high, and $E_{predicted}$ is dependent mainly on catchment slope. This

results in values clustering at NSE $\sim 0.34$ (Fig. 4c, d, e) for models with high diffusion. Because $D/K$

covaries with $n$ (and $m$, Fig. 4a), we find that sediment transport derived from diffusional processes is

maximised when $E_{predicted,diffusive}/E_{total}$ is $\sim 0.43$ (Fig. 4f).

### 4 Discussion

In their benchmark study, Portenga and Bierman (2011) employ stepwise regression to relate their

compilation of [10]Be-derived denudation rates to a range of factors embracing topography, climate,

lithology and seismicity. That study, along with the later inclusion of normalised steepness (e.g., Harel et al., 2016; Marder and Gallen, 2022), added substantially to our knowledge of how and why denudation rate varies. Our alternative approach here focuses upon the erosional processes at play in

terms of advective and diffusive mass flux, rather than attempting to interpret the machinations of landscape response to internal and external agents. A significant advantage is that explicit relations between $m$, $n$, topography, denudation, and LEM parameters are derived at the scale of the DEM grid cell within each catchment, and success is gauged from the absolute difference between modelled ($E^*_{predicted}$) and [10]Be-derived ($E_{apparent}$) denudation rates. In other words, we evaluate LEM parameters as

they are commonly implemented in the models.

**4.1 Optimised parameters for landscape evolution models**

We first consider some comparisons with previous work regarding advection-only approaches. Our optimised $A_c \sim 0.05$ km$^2$ (Fig. 2b) for the 3-arc-sec resolution models falls near the minimum of the

range applied in previous studies, such as Whipple and Tucker (1999), who suggest 0.059–0.14 km$^2$. Our optimised $n \sim 1.3$ (n ~ 1.4 for the 1-arc-sec models) is much lower than the 2.6 reported by Harel et al. (2016), which is derived from regression of denudation rate and normalised steepness ($k_{s\,ref}$). Harel et al. (2016) then use the product of $k_{s\,ref}$ and a scaling drainage area to calculate $M_\chi$. In principle, $M_\chi$ and $n$ should be similar to our $K$ and $n$ values; however, $n$ is derived from a regression of $E_{apparent}$

against $M_\chi$ and it is integrated based on each pixel within the catchment. The large discrepancy between our globally optimised values of $n$ and those of Harel et al. (2016) may stem from the inability of the latter method to accommodate inherent nonlinearities at the sub-catchment or sub-reach scale when $n \neq 1$, whereas our approach is designed to capture some of these nonlinear effects. This is particularly important in transient catchments, as spatial heterogeneity in denudation rate is

often controlled by steep areas in the catchment, such as knickpoints, and higher values of $n$ amplify the proportion of denudation derived from steep areas relative to the rest of the catchment (Fig. 1).

While linear diffusion ($p = 1$) is commonly applied in landscape evolution studies (e.g., Forte et al., 2016), our optimised $p \sim 2$ for the diffusion-only model is consistent with Gabet al. (2021) in

which denudation rate correlates best with the square of hillslope convexity. In response to Gabet et al.

(2021), Struble and Roering (2021) point to a systematic underestimation of curvature in natural

landscapes that may be an artefact of the numerical methods used for estimating curvature from

DEMs. Gabet et al. (2021) employ high-resolution (~1 m) LIDAR data, but the broader point made by

Struble and Roering (2021) poses a serious limitation for large-scale LEM analyses that are typically

restricted to lower-resolution DEMs. In such cases, the need for mass conservation and numerical

stability are important considerations. And yet, a diffusion equation with exponent $p \neq 1$ is

numerically unstable, physically unexplained, and does not accommodate deposition (the result of

negative curvature). What does it say about the utility of running LEMs on natural landscapes if the

optimised parameter value ($p \sim 2$) cannot be implemented? Struble and Roering (2021) suggest that $p$

$\sim 2$ enhances the influence of steep, rapidly eroding areas on average curvature, which are commonly

underestimated by many methods. Below, we discuss how the influence of these steep areas may be

approximated by the stream power equation coupled with linear diffusion.

Our advection-diffusion model allows us to explore aspects of how hillslope and river

processes govern sediment flux in river catchments. Theoretically, for a catchment in a perfect state of

mass-flux equilibrium (or steady state), hillslopes and rivers are eroding at the same rate, so either one

should be equally useful as a proxy for denudation rate. There would be no apparent advantage to

combining advection and diffusion in the same model since they would both yield the same average

denudation rate. Landscapes are, however, more often not at steady state (at least over timescales

integrated by cosmogenic [10]Be) and the slight dominance of advective denudation in our optimised

model ($E_{predicted,diffusive}/E_{total} = 0.43$, in Fig. 4f) suggests that transient signals disproportionately affect

catchment-averaged denudation rates.

The positive relationship we observe between optimised $n$ and relative diffusivity gives rise to a

compelling possibility: as diffusivity increases, advective denudation becomes less important as a

proxy for the total average denudation rate within the catchment, and more of a proxy for transience

focused at the most rapidly eroding zones. However, in the absence of diffusion, river incision must

account for all sediment that would be otherwise eroded diffusively from hillslopes. The increase in

optimised $n$ with diffusivity therefore represents the expanding role of steep transient zones in dictating the catchment-scale denudation rate. Our optimised result for the advection-diffusion model, $n \sim 2.3$ (Fig. 4b), is compatible with previous work suggesting that typically, $n > 1$ (e.g., Lague, 2014; Harel et al., 2016).

The best correlation between predicted and apparent denudation rates occurs when $D/K \sim 1.79 \times 10^6$ (Fig. 4d). This outcome broadly agrees with other studies that use $K$ values in the range $\sim 10^{-8}$ to $10^{-5}$ m$^{(n-1)}$ yr$^{-1}$ and $D$ values in the expected range noted by Fernandes and Dietrich (1997) of $4.4 \times 10^{-4}$ to $3.6 \times 10^{-2}$ m$^2$ yr$^{-1}$. Whipple et al. (2017) reports an optimal $D/K$ ratio of $5 \times 10^2$ from Himalayan catchments, although fixing $n = 1$ in their models is a limiting assumption because $D/K$ covaries with

$n$, as we show. The diffusion model employed here assumes that the long-term flux of hillslope material is similar to the amount transported in one time-step. In reality, catchment may not be at steady state and the hillslope denudation rate may change notably so as to change the rate of hillslope flux within individual catchments.

## 4.2 Erosion and precipitation

Correlating topographically-derived metrics with mean annual precipitation (MAP) on a global scale has been a long-standing goal (e.g., Ahnert, 1970). Harel et al. (2016) examine correlations between stream-power variables and climate as defined by the Köppen-Geiger scheme. They find that aridity yields the lowest $n$ and that, in general, $k_{s\,ref}$ and temperature covary inversely: warm deserts yield the

highest $k_{s\,ref}$ and polar regions the lowest, on average, albeit with large uncertainties. Because our approach compares model results to denudation rates, rather than using regression, we can more directly correlate $K$ and $n$ as they are implemented in LEMs under differing climate. This also means that our method is potentially better suited for transient catchments.

     We demonstrate a general increase in $n$ and $p$ with precipitation for both the advection model and the

diffusion model (Fig. 5b, c), with the exception of the highest MAP bin ($\sim 2300$–$6500$ mm yr$^{-1}$). Marder and Gallen (2022) also observe a levelling-off in $n$ for the wettest environments along with a relatively lower correlation of $K_{sn}$ vs erosion rate in these wetter environments compared to other environments. Our advection-

diffusion model shows a similar increase in *n*, aside from the lowest MAP bin (Fig. 5d). In the stream-power based models, the rising *n* with respect to precipitation suggests that wetter environments favour a nonlinear erosional response perhaps tied to heavy-tailed flood-frequency distributions. Extremely variable flow regime is characteristic of drylands (Zaman et al., 2012); and yet, in two of the three models (advection-only and diffusion-only), the exponent (*n* or *p*) is low. By contrast, only the advection-diffusion model yields a relatively high *n* (~ 2.8) for drylands. This may indicate that the advection-diffusion model, which we suggest is more sensitive to transience, does a better job at simulating dryland catchments. However, we emphasize the relatively low performance of all models (NSE = 0.2–0.3) for the driest settings (Fig. 5b-d), which may reflect the challenge of capturing the erosional dynamics of drylands with such simple models.

It is difficult to differentiate changes in exponents from changes in model coefficients, due to their inherent coupling in power-law functions (e.g., Syvitski et al., 2000). In response to this issue, we calculate coefficients *D* (for diffusion-only) or *K* (advection) for each of the optimised models (i.e., with constant exponents) using Eq. (*5*) in catchments represented by twenty MAP bins distributed regularly (N = 182±1; Fig. 5e). By looking at how the coefficients vary within each bin, we effectively reduce the influence of topography and isolate the relationship between erosion rate and precipitation. We emphasize that, due to the wide range of data used, optimized models can have large uncertainties (dry regions having the highest uncertainty relative to the mean) with residuals that are generally log-normally distributed (Supplementary fig. S8). We also trialled different drainage-area bin sizes, where a similar (albeit noisier) pattern remains with higher bin density (Supplementary fig. S7).

Optimised coefficients show a local peak in denudation rates centred around 300 mm yr$^{-1}$ and then dipping overall from around 1100 to 1600 mm yr$^{-1}$ before increasing again for extremely wet regions (Fig. 5e). These results agree well with the classic work of Langbein and Schumm (1958), which suggests that the fastest eroding environments are semi-arid (MAP ~ 250 mm yr$^{-1}$). This relationship is thought to be a product of the interplay between denudation, vegetation, total precipitation, and storm frequency in semi-arid regions—an outcome reproduced by Istanbulluoglu and Bras (2006), who show a positive relationship between sediment transport and the effects of reduced sediment cover/increased runoff during drought.

While Langbein and Schumm (1958) had scant access to data from wetter settings, our results reveal an upward trend in coefficient values for MAP > 1500 mm yr$^{-1}$ (Fig. 5e). This is in line with Walling and Kleo's (1979) global study of sediment yield and climate, which also shows a further major peak at ~ 800 mm yr$^{-1}$ and may be attributed to the most expansive agricultural production globally (e.g., Hyman et al., 2016). In contrast to Walling and Kleo (1979), which does not isolate the

effects of variable land use, topography, and geology on sediment yields, our use of denudation rates based on [10]Be means that we can largely ignore the effects of land use. This may explain the subdued peak ~ 800 mm yr$^{-1}$ in our data.

       Marder and Gallen (2022) find a nonlinear relationship between [10]Be-derived denudation rate and $k_{sn}$ via regression; they also find that the exponent relating $k_{sn}$ to erosion rate (here effectively

equivalent to $n$) increases with precipitation. They took steps to exclude transient catchments, as transience may influence $n$ due to the non-linear effects of stream power, as noted above. Likewise, we find $n$ generally increases with precipitation, giving credence to the idea that our approach may account to some degree for transience. In particular, we envisage cases in which steep zones, such as adjusting areas downstream of knickpoints, are responsible for a large proportion of the total

catchment erosional flux, even when they may represent a small fraction of the drainage area (Willenbring et al., 2013). Where denudation is a nonlinear function of slope and drainage area, an average of these denudation rates is unlikely to be proportional to the integrated $k_{sn}$.

       One acknowledged shortcoming of our approach (noted in Section 2.4) is that some lithologies may be over-represented in areas with higher or lower MAP. As we discuss in the next section, it is a

challenge to discriminate biases due to lithology from those linked to precipitation in bins that span heterogeneous lithologies.

### 4.3 Erosion and lithology

We assessed the variability of LEM parameters within each lithological bin following a similar

approach to Section 4.2, with the additional step of employing $K_{lith}$ (Eq. 9) to isolate the effects of lithology on $K$ in the advection-only model. The three-fold range in erodibility (Fig. 6e) is much lower

than that reported elsewhere, in some cases by several orders of magnitude (Sklar and Dietrich, 2001; Garcia-Castellanos et al., 2018). This may be due to the greater focus on the differential erodibility within individual sites (Garcia-Castellanos et al., 2018). Despite our efforts to account for some of the covariation with MAP, our analysis inevitably smooths out some variability owing to the diversity of catchments incorporated within each lithological bin.

We find that for the advection-only model (Fig. 6b), $n$ is higher in metamorphic and plutonic-intermediate rocks, which also tend to be more resistant (e.g., Moosdorf et al., 2018). This is not surprising given that $n$ is thought to be influenced by higher thresholds for rock detachment, which in turn can lead to more non-linear behaviour. More complex relationships involving hillslope diffusion may result in the general lack of a comprehensible pattern emerging in the advection-diffusion models (Figure 6c & 6d). The higher number of parameters used in the advection-diffusion model leads to broader distributions (Fig. 6d).

When looking at the variability in $K$, our overall results are somewhat expected: all three models agree on the general tendency of sedimentary rocks being most erodible and plutonic/volcanic being least, although the relative magnitudes of these differences vary between models. Unconsolidated sedimentary rock is the first or second-most erodible of all according to the diffusion and advection-only models (Fig. 6e). Models disagree about the least erodible subcategories: basic plutonic, intermediate volcanic, or pyroclastic. The pyroclastic, plutonic intrusive, and unconsolidated sediment rock-types show the most variance. For pyroclastic, the advection models suggest relatively low erodibility, whereas the diffusion-only models suggest moderate erodibility. Most models agree on the moderate erodibility of plutonic intrusive rocks, but due to the low sample size, the uncertainty is very high. Less expected is the relatively high erodibility of carbonate sedimentary rocks shown for all our models. For example, other studies argue that carbonates should be less erodible (Ott et al., 2021), and that volcanic rocks may be relatively more erodible than we have shown here (Moosdorf et al., 2018). Some of these variable findings possibly arise due to factors, such as fracture density and weathering condition, which are difficult to accommodate in large-scale analyses (Neely et al., 2019), and also contribute the large uncertainty (Supplementary fig. S9).

### 4.4 Erosion and drainage area

Are LEM parameters sensitive to catchment drainage area? Violin plots (Fig. 7) show exponents ($n$ and $p$) values binned according to drainage area. The advection-diffusion model shows no dependence on area, but for the advection-only and diffusion-only model, the exponent values generally increase with drainage area, $n$ ranging from ~1.1 to ~1.5, and $p$ ranging from ~ 1.8 to 2.2. Additionally, we note the extreme range in $n$ (~ 1.3 to > 3) for small catchments in the advection-diffusion models, potentially indicating the influence of landslides in headwaters (e.g., Yanites et al., 2010). This dependence of $n$ and $p$ on drainage area may be difficult to discern from other effects, but one possibility may be the inherent link with precipitation. Larger catchments are more likely to cross large rainfall gradients, which as we have shown, gives rise to higher $n$ or $p$.

### 4.5 Limitations and future considerations

An important limitation of our approach is that it fails to unravel the $m/n$ ratio, which is known to vary widely (Gailleton et al., 2021). We ran several trials (using optimized $A_c = 0.05$ km$^2$ and 20 values of $n$ linearly spaced from 0 to 4) and found that $m/n = 0.3$, 0.45 and 0.6 yielded negligible difference in fit (NSE = 0.48, 0.47, and 0.45, respectively) despite the similar values of optimized $n = 1.4, 1.2$, and 1.2, respectively (Supplementary fig. S10). One option for improvement is to determine $m/n$ from the river profile concavity using recent integral-based techniques (e.g., Harel et al., 2016, Gailleton et al., 2021) before running the suite of models. This may be especially useful for tuning the $m/n$ ratio within specific regions.

Our approach cannot account for several factors that are known to bias [10]Be-derived denudation rates (e.g., von Blanckenburg, 2005; Dingle et al., 2018; Hippe et al., 2019; Struck et al., 2019). Such factors violate two key underpinning premises of the method: steady and uniform erosion across the catchment and continuous exposure of sediment at/near the surface (noted in Section 1.1). For example, sudden pulses of sediment from sources of deep-seated mass wasting can shift apparent denudation rates downward by diluting the nuclide abundances in the river sediment sample. Given

that our LEM is deterministic, it is difficult to model the impact of landslides on nuclide inventories, particularly in small headwater catchments where their influence will be greater (e.g., Yanites et al., 2010). A second pertinent issue is that our modelling assumes the erosional flux is transported instantly to the catchment outlet without intermediate storage. And yet, river sediment is likely to experience multiple episodes of erosion and deposition, especially in large lowland catchments where the volume of sediment storage expands greatly. Intermediate sediment storage can push [10]Be-derived denudation rates either upwards or downwards depending on the duration and depth of sediment burial (Wittmann and von Blanckenburg, 2016; Struck et al., 2018). The extent of bias in the *OCTOPUS* dataset (Codilean et al., 2022) is essentially unknown but it no doubt contributes to the data scatter we report here.

In addition to the complications with stream-power and diffusion-based models (noted in the Introduction), the inherent assumption that channel width increases monotonically with stream discharge (or its proxy, drainage area) is one that is violated widely. Channel width narrows at knickpoints (Whitbread et al., 2015; Yanites et al., 2018) and across diverse substrate erodibilities (Jansen et al., 2010; Croissant et al., 2017). The value of *n* extracted from transient landscapes will be influenced by this process, and a goal of future global analyses should be to incorporate more sophisticated rules for channel width evolution (e.g., Yanites et al., 2018).

The approach described here is sufficiently robust to incorporate many different types of models. We would like to see, for instance, an exploration of the way in which drainage is routed through catchments. Different flow routing schemes can give rise to notably different drainage networks and corresponding drainage areas (e.g., Endreny et al., 2003). Here we have used unidirectional 'D8' flow, but alternatives such as multiple flow routing can produce alternative results and perspectives (e.g., Pelletier, 2004). Future efforts may test a range of flow routing methods against denudation rate in order to test their efficacy.

## 5. Conclusions

We have examined the most widely used parameters applied to a set of three landscape evolution model set ups: (1) a stream-power based, advection-only model; (2) a diffusion-only model; and (3) an advection-diffusion hybrid model. We optimised the parameter values by comparing directly the catchment-averaged denudation rates predicted by our three models with a global catalogue of [10]Be-derived apparent denudation rates (Codilean et al., 2022).

The diffusion-only model outperformed the advection-only model when applying $p \sim 2$. However, the physical implications and numerical limitations of this result make it impractical for implementation in LEMs. Instead, we propose that linear diffusion coupled with fluvial denudation (advection-diffusion) captures a high proportion of sediment derived from rapidly eroding, steep areas in a similar sense to a diffusion model with exponent $p \sim 2$. In the advection-diffusion hybrid model, the best agreement between the predicted and apparent denudation rates is observed with $n \sim 2.3$ (assuming a fixed concavity, $m/n = 0.45$), while the ratio of diffusivity/advection coefficient ($D/K$) is optimised at $\sim 1.79 \times 10^6$.

The Monte Carlo method employed here is a simple and powerful means of identifying ideal parameter sets over large spatial scales and is especially useful for dealing with sparse datasets. We applied the same approach to elucidate differences in optimal LEM parameters when considering lithology and precipitation. By looking at the LEM coefficients, we were able to better account for the influence of topography when isolating the relationship between denudation rate and precipitation/ lithology. Of particular interest was a general upward trend in the coefficients ($K$ and $D$) with respect to precipitation, and a local maximum centred at $\sim 300$ mm yr$^{-1}$. This local maximum may represent the higher erodibility of semi-arid environments identified by Langbein and Schumm (1958).

Nevertheless, many other influences on denudation are yet to be explored in a robust way. Future studies may use these parameter ranges as a baseline to inform large-scale landscape evolution studies. Moreover, our methodology could be extended to incorporate more complexity into the canonical advection and diffusion-based equations applied here.

**Author contributions:** GR conceived the study, performed data analysis, and devised the code. JDJ and PV assisted with framing the study, and LYM performed code analysis. All co-authors contributed to MS production.

**Resource availability:** The functions and notebooks for running this analysis are available from www.github.com/ruetg/lem_global_optimize.

**Acknowledgments**: Many thanks to Bruce Wilkinson for his guidance in framing the study, and to Daniel Garcia-Castellanos and Kim Huppert for their constructive advice. We are grateful for reviews from Richard Ott and Boris Gailleton, who encouraged us to explore new datasets and methods and greatly improved the MS. Several calculations were performed on the CSDMS Blanca HPC at the University of Colorado, Boulder. Maps in Figure 1 were produced using PyGMT (pygmt.org).

**Competing Interest:** Authors declare no competing interests

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

**Tables**

**Table 1:** Parameter values and ranges for the three model set ups.

| Model set | Parameter | Range | Sampling |
|---|---|---|---|
| Stream power only | $n$ | 0–4 | Random, 1000 samples |
| | $A_c$ | 0.01–8 km$^2$ | Random, log-uniform, 1000 samples |
| Diffusion only | $p$ | 0–4 | Linear increment by 0.2 |
| Stream power + diffusion | $n$ | 0–4 | Random, 10,000 samples |
| | $D/K$ | $10^3$–$10^{10}\ m^{0.9n+1}$ | Random, log-uniform, 10,000 samples |
| | $A_c$ | 0.01–8 km$^2$ | Random, log-uniform, 10,000 samples |

**Figures**

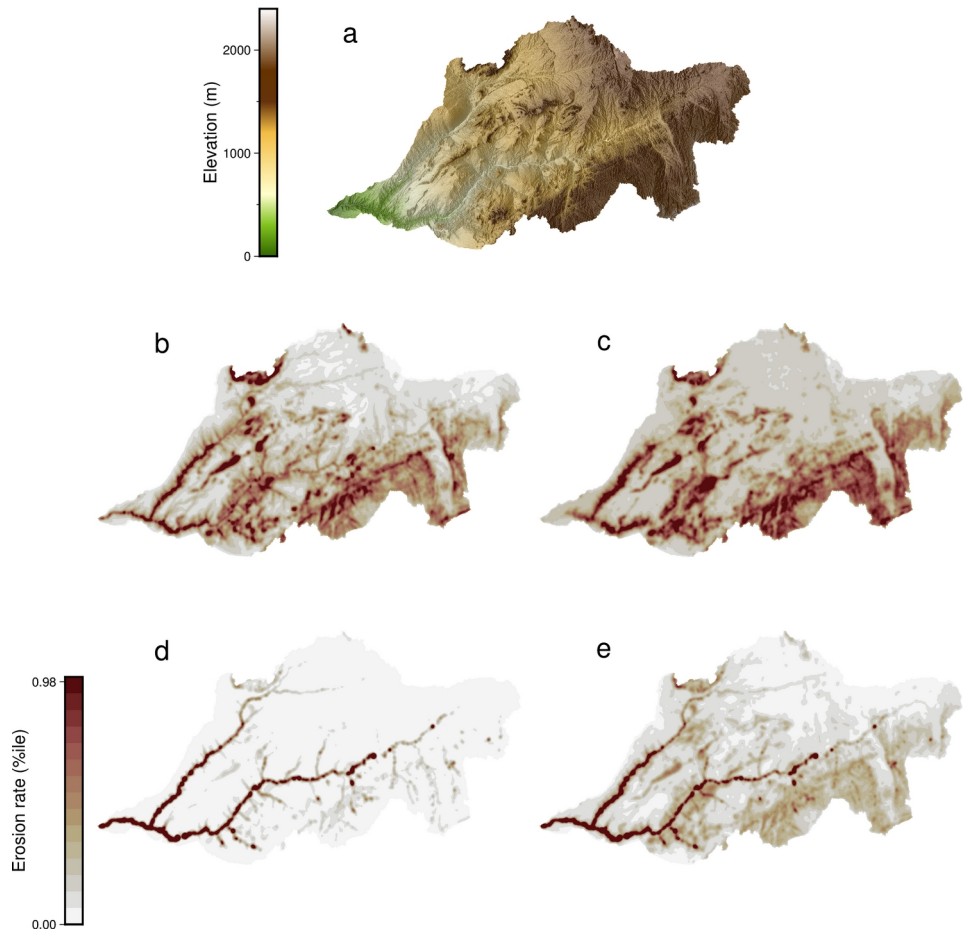

**Figure 1:** a) Catchment example (Swakop River, Namibia) clipped from a *Hydrosheds* DEM based on the shapefile provided in *OCTOPUS* v.2 (Codilean et al., 2022). Lower panels (b–e) show corresponding relative denudation rates (colour ramp spans 0–98 % of the range) for differing parameter values. No diffusion is included in (b) and (d), hence erosion is focused in the channels. In (c) and (e), a moderately high ($10^7$) diffusivity is used relative to advection, which causes erosion to be more focused on hillslopes.

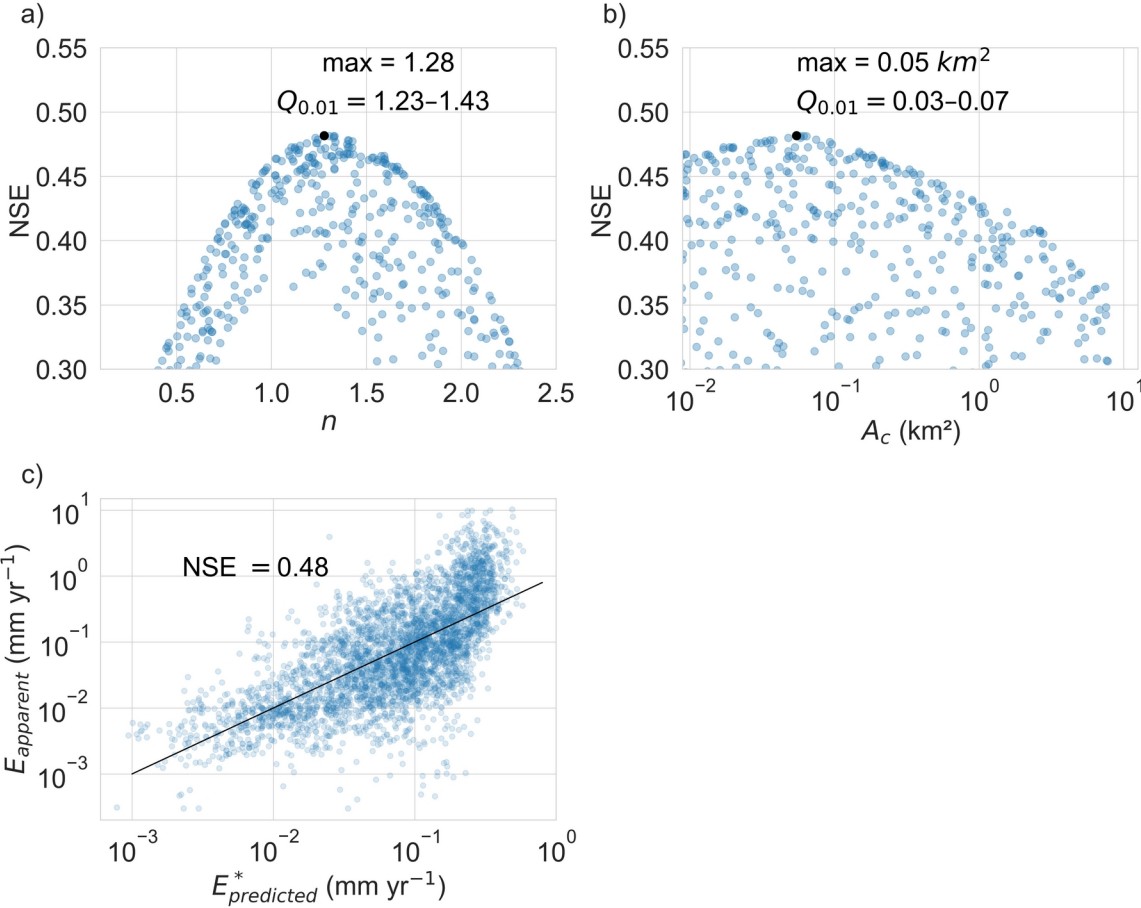

**Figure 2:** The advection-only model. a) Optimised $n$ = 1.28 ($Q_{0.01}$ = 1.23–1.43), b) Optimised $A_c$ = 0.05 km² ($Q_{0.01}$ = 0.03–0.07 km²), c) Apparent vs predicted erosion rate, NSE = 0.48; no regression is performed, the black line indicates a 1:1 fit.

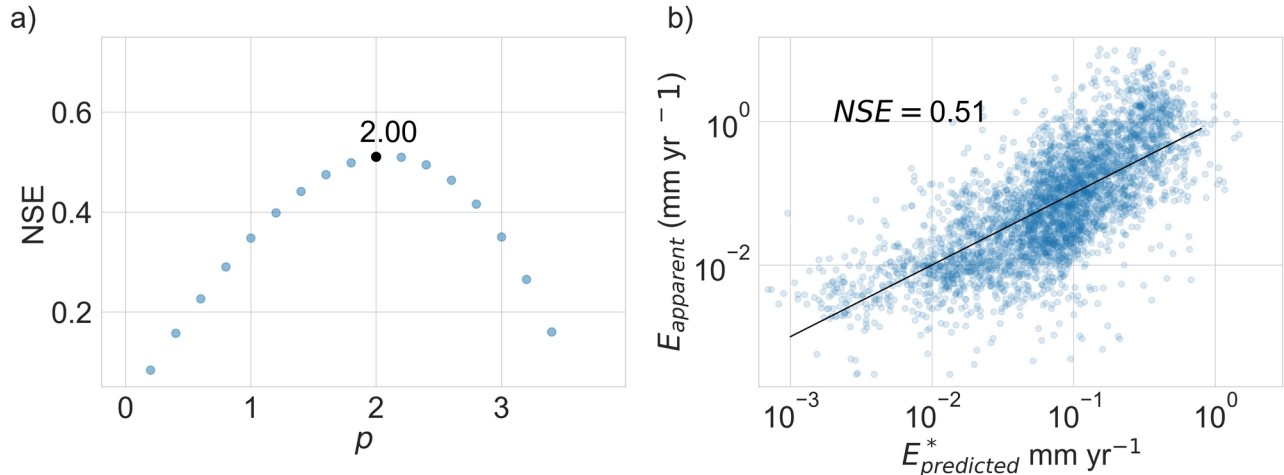

**Figure 3:** The diffusion-only model. a) The sole free parameter ($p$) is optimised at $p = 2.00$. b) Apparent vs predicted erosion rate, NSE = 0.51. The Black line represents a perfect 1:1 fit.

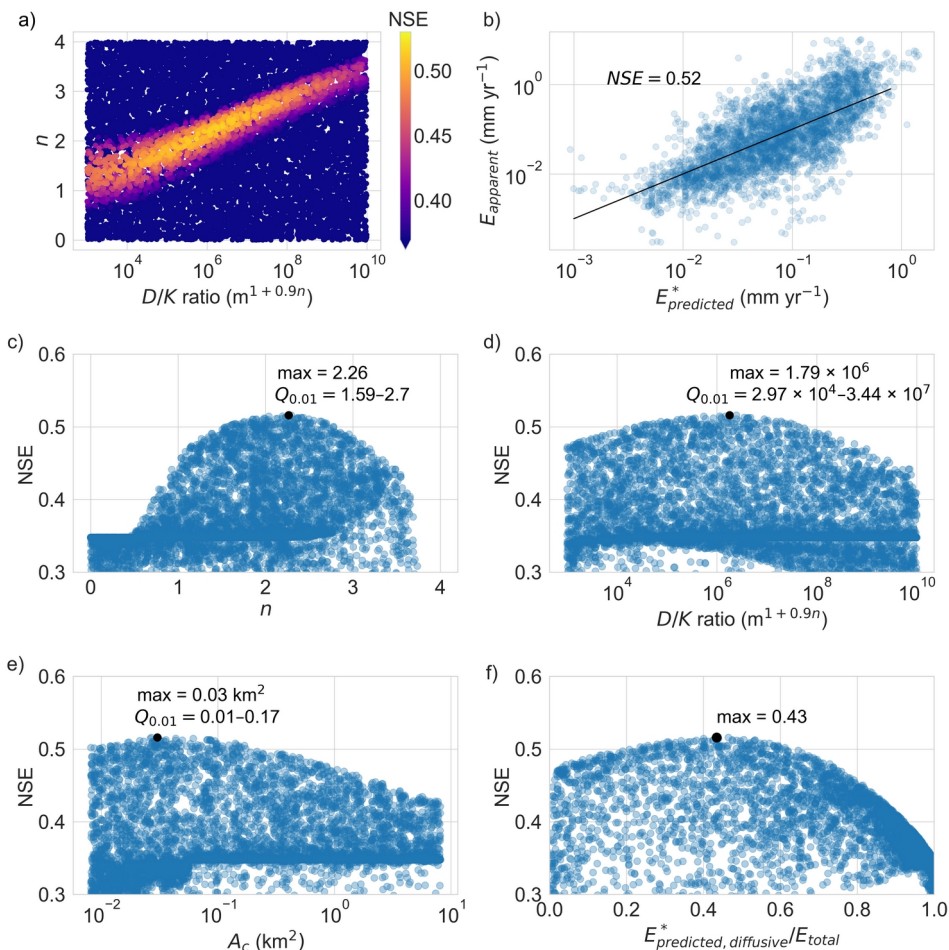

**Figure 4:** Model parameters representing variations in the relative dominance of advection vs diffusion. a) Covariance of $D/K$ with $n$; when $D/K$ is low (no diffusion), optimal $n$ approaches ~1.3 (y intercept). b) The best correspondence between $E^*_{predicted}$ and $E_{apparent}$ is achieved with NSE = 0.52, where c) $n \sim 2.3$ ($Q_{0.01}$ = 1.59–2.7); d) $D/K \sim 1.79 \times 10^6$ ($Q_{0.01} = 2.97 \times 10^4$ to $3.44 \times 10^7$), and e) $A_c \sim 0.03$ km$^2$ ($Q_{0.01}$ = 0.01–0.17 km$^2$). Clustering at NSE ~ 0.34 in panels (c, d, e) represents parameter sets where diffusion dominates over advection. f) Sediment transport derived from diffusional processes is maximised when $E_{predicted,diffusive}/E_{total}$ is ~ 0.43.

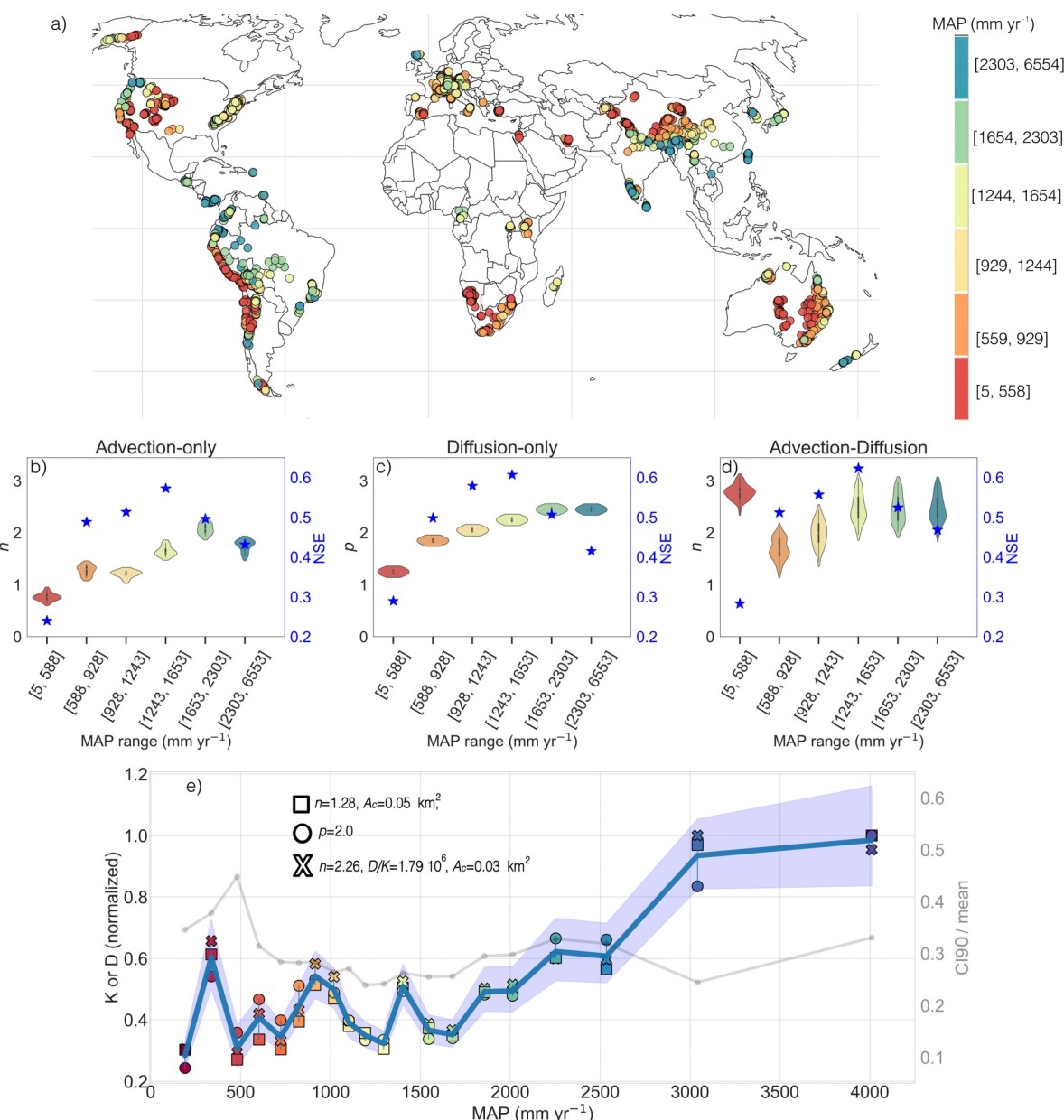

**Figure 5:** a) Global distribution of catchments in the *OCTOPUS* v.2 catalogue of [10]Be-derived apparent denudation rates (Codilean et al., 2022) coloured by mean annual precipitation (MAP). b-d) Violin plots (symmetric kernel density plots) of $n$ or $p$ values within the top 1% of model runs for the advection (b), diffusion (c), and advection-diffusion (d) models. Blue stars correspond to the NSE value for each bin. e) Coefficients for diffusion (circle), advection (square), and advection-diffusion (cross) calculated for each globally optimised model per MAP bin. Diffusion model plots represent only the global maximum within each bin, due to insufficient model runs to form a distribution. All panels use the same colour ramp, which corresponds to the MAP bin; blue shading represents the average bootstrapped 90% CI spanned by the 3 models, and heavy blue-line is the mean. The grey line represents the range spanned by the 90% confidence interval averaged among all 3 models relative to the mean (see Supplementary fig. S8).

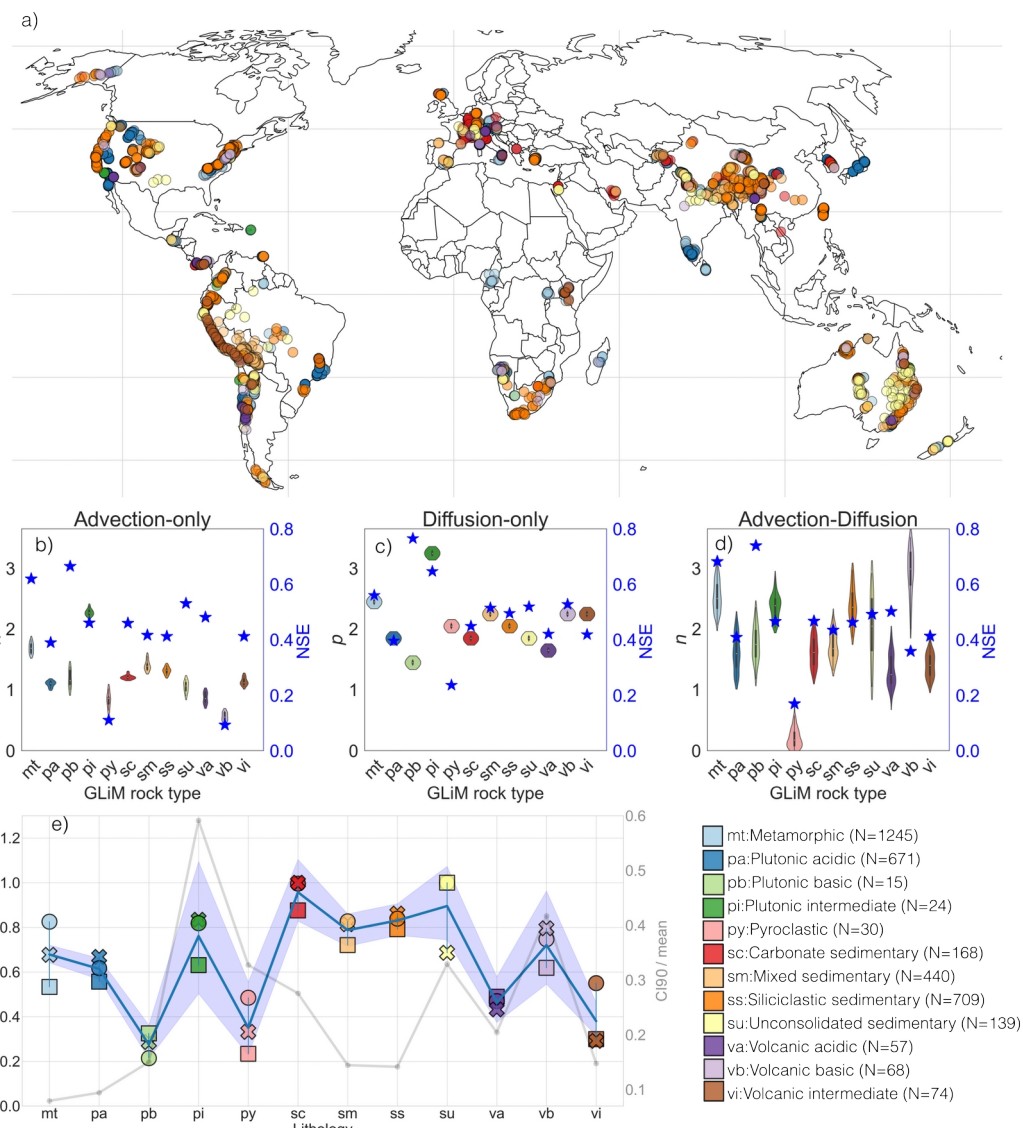

820

**Figure 6:** a) Global distribution of catchments in the *OCTOPUS* v.2 catalogue of $^{10}$Be-derived apparent denudation rates (Codilean et al., 2022) coloured by dominant lithology. b-d) Violin plots of *n* or *p* values within the top 1% of model runs for the advection (b), diffusion (c), and advection-diffusion (d) models. Blue stars correspond to the NSE for each bin. e) Coefficients (normalised by their maximum values) for different best-fit models within 12 lithologic subsets (from Hartmann and Moosdorf, 2012). Coefficients for diffusion (circle), advection (square), and advection-diffusion (cross) calculated for each globally optimised model per lithologic bin; blue shading represents avg. bootstrapped 90% CI spanned by the 3 models, and heavy blue-line is the mean. All panels use the same colour ramp, which corresponds to the lithologic bin. The grey line represents the range spanned by the 90% confidence averaged among all 3 models relative to the mean (see Supplementary fig. S9).

835

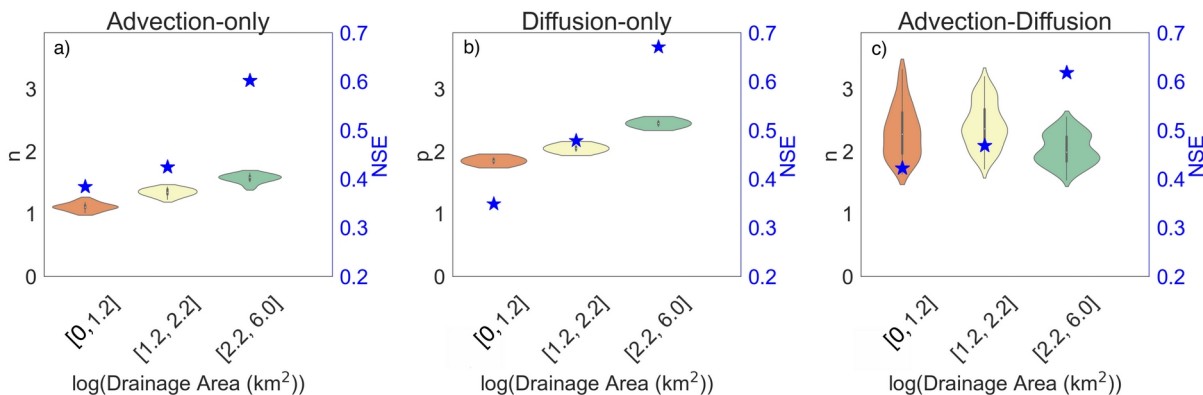

**Figure 7**: Distributions of *n* or *p* within each area-bin for (a) advection-only, (b) diffusion-only, and c) advection-diffusion models. Blue stars represent the corresponding NSE for each bin. The top 1 % of model runs are used.