# Peer review of "Optimising global landscape evolution models with 10Be"

_Earth Surface Dynamics, 2022_

## Author Comment (AC1)

**Overview**

We thank the reviewers for their excellent points and supportive comments. We have made a major overhaul both with regard to the text and the modelling.

**Modelling**: We have included a smoothing algorithm as recommended by both reviewers, and we have compared our results using SRTM with those of the Copernicus 30m DEM. We have performed sensitivity tests comparing models under different ratios of *m/n*. We have included a new dataset for binning, CHELSA for MAP. As a result of some of these models, some of our optimized parameter values have changed modestly, but the main findings remain largely unchanged. As this has increased the number of supporting figures, we have opted to place most of them in a file separate from the main text.

**Discussion**: We have added more analysis exploring the relationship between LEM parameters and Lithology, Precipitation, and (now) basin size along with their variability, in line with comments made by both reviewers. We have added a significant amount of text related to the limitations of this approach and comparing it to the results of previous authors.

This represented a large task, and we are confident that it has greatly improved the results and impact of the paper.

This study by Ruetenik et al. investigates parameters of diffusion and advection models for landscape evolution, based on a global compilation of CRN erosion rates. The authors use CRN data to optimize parameters in their landscape evolution models and investigate the distribution of parameters in respect to the different models, as well as environmental variables of precipitation and lithology. The authors advertise their method as a way to more objectively select parameter values for landscape evolution models (LEMs). I believe this study is of great interest for the geomorphological community and is generally suitable for Esurf. The parameter optimization results will be of value to justify model parameter selection and offer insights into the processes driving landscape evolution. Furthermore, the dependencies of coefficients on climate and lithology have been widely discussed, and this paper offers some valuable insights here. However, I have several points concern that should be addressed before publication.

The current approach incorporates catchments with a huge variation in drainage area. This may lead to a lot of different biases, which arise from the CRN erosion rate assumptions. I would like to see, how the results look like if you only use 10-1000km² catchments, or similar. Small catchments are prone to biases of sudden sediment input or anthropogenic disturbance. Large catchments (> 500km²) typically violate the uniform quartz fertility and sediment contribution assumptions, as well as having a negligible transport time without nuclide buildup. Also, production rate uncertainties grow significantly for large catchments due to the before mentioned reasons. I understand that this will substantially decrease the quantity of data points, but might on the other hand substantially increase the data quality.

Thank you for this point. We did incorporate <10,000 $km^2$ catchments in the analysis for Appendix A of the original MS, but this was primarily due to the limitations of the higher resolution DEMs and not intended as a direct comparison. We have decided to include another section in the discussion where we bin parameters based on drainage area as well in an attempt to address this point. This is now contained in section 4.4.

L476: "Are LEM parameters sensitive to catchment drainage area? Violin plots (Fig. 7) show exponents (n and p) values binned according to drainage area. The advection-diffusion model shows no dependence on area, but for the advection-only and diffusion-only model, the exponent values generally increase with drainage area, n ranging from ~1.1 to ~1.5, and p ranging from ~ 1.8 to 2.2. Additionally, we note the extreme range in n (~ 1.3 to > 3) for small catchments in the advection-diffusion models, potentially indicating the influence of landslides in headwaters (e.g., Yanites et al., 2010). This dependence of n and p on drainage area may be difficult to discern from other effects, but one possibility may be the inherent link with precipitation. Larger catchments are more likely to cross large rainfall gradients, which as we have shown, gives rise to higher n or p."

No smoothing of the landscape is mentioned. Flow paths from DEMs especially in valley bottoms tend to have large errors. These errors typically lead to a lot of slope values = 0 and a bunch of very high values (Schwanghart and Scherler, 2017). This would bias the predicted erosion rates, if no smoothing is applied before running the LEM. I hope this was addressed and not mentioned, otherwise smoothing should be applied to the flow paths and the method described.

This is a fair point – we did not include smoothing in the original analysis, only the lake filling algorithm, as this is the most common approach in LEMs.  However, recognizing the potential for bias on real DEMs, we have re-run the advection and advection-diffusion models and we now use a smoothing algorithm on the stream profiles which modified the results only slightly.

Details:

 Although we found that the algorithm of Schwanghart and Scherler, 2016 was was not computationally feasible for many of the larger catchments, we were able to apply a  ~ 1km averaging window on the landscape which appears fairly effective at most scales (see figure below). Interestingly, however, the smoothing seems to have had very little effect on the SP-only results, which was also somewhat of a surprise as I also would have thought that the larger flat regions and exaggerated steep zones would have had somewhat of an effect.  I think this may demonstrate that the larger steep regions (not necessarily the smaller-scale <1km noise)  have the largest influence on erosion rates. Specifically, the advection only model results are essentially unchanged, while the optimized advection-diffusion coefficient is higher (2.26) which as we mentioned is likely due to the increased sensitivity of *n* to perturbations (noise or otherwise) in the presence of diffusion.

[Figure]

Supplemental Figure for reviewers: Results of a ~1 km smoothing algorithm shown in orange, with the original profile in blue.

I like the interpretation put forward to explain the variation in coefficients with precipitation (MAP). However, without further analysis this should be stated as speculation and probably reduced in text. The problem I see is that variations in lithology with MAP are not accounted for. This means that if the catchment lithologies are not homogeneously distributed among climate zones, which can happen easily due to the high clustering of CRN measurements in certain regions, one could get significant biases. Either an analysis of MAP values with respect to the lithologies would need to be added, to show that the distribution is homogeneous, or this caveat needs to be clearly acknowledged.

Thank you for this point. In response to your comment and an email we received from another researcher, we added new discussion to clarify the relationship of our results with respect to Marder and Gallen, 2022. This also relates in part to our response to the review by Dr. Gailleton, where we have further analysed the individual bins with respect to $n$ to find a similar increase in $n$ with respect to precipitation. However, while Marder and Gallen 2022 selected specifically for steady state basins, our results include transient basins, potentially demonstrating the utility of the method even in non-steady state conditions. However, we also emphasize the inherent link between coefficients and $n$, which we are not able to fully resolve. Also, with the new CHELSA data the trend is similar, but the ~300 mm/yr peak is slightly smaller and the 800 mm/yr peak is larger. Overall, we acknowledge the limitations brought forth by a potential coupling between lithology and climate now in the discussion:

L439: "One acknowledged shortcoming of our approach (noted in Section 2.4) is that some lithologies may be over-represented in areas with higher or lower MAP. As we discuss in the next section, it is a challenge to discriminate biases due to lithology from those linked to precipitation in bins that span heterogeneous lithologies."

L264: "Precipitation differences between lithologic subgroups can be significant; for instance, averages range from 851 to 2077 mm yr-1 for unconsolidated sediment and metamorphic rocks, respectively"

Line 10-12: It should be mentioned that many LEMs simulate erosion processes with the Stream Power Incision Model (SPIM) and therefore need these parameters as input. It should not necessarily be assumed that all LEMs run this way, because there are also transport-limited or combined approaches.

This is a good point and we have clarified this on line 12, that we are referring to stream-power based models

Line 22: Somewhat outperforms? Please, be specific.

Good point we have clarified that we are referring to a modest correlation increase.

Line 73: Denudation is more correct than erosion in most cases. However, if you were to be strict, neither of the two terms would be correct (e.g. due to mass loss below the attenuation length of cosmogenic nuclides). The best strategy could be to have a very brief definition of what is meant by denudation or erosion, describe why a certain term is used, and then be consistent, and also remove the parenthesis in the abstract.

We have changed relevant instances of "erosion" to "denudation" added the definition on L52.

"Note that we use the term 'denudation' to denote the mass loss leading to lowering of Earth's surface"

Line 79-81: This is more of a comment. It makes me slightly uncomfortable that two of the main assumptions are being highlighted here, for reasons that are not apparent to me, and other important assumptions (quartz fertility, uniform contribution of stream sediment proportionally to local denudation rate and area) are being folded away into the next sentence.

Agreed that it was a little misleading to emphasize those first two points above the rest. We have slightly modified these sentences (now starting on L94) so that it reads as a list of example assumptions.

Line 107: There are more processes that can affect the value of $n$. For instance, incision process (Whipple et al., 2000), but also other flow resistance parameters, or other processes creating incision thresholds (Lague et al., 2005).

We have specified the incision threshold dependence, and added a discussion on this related to lithology in section 4.3.

L453: "We find that for the advection-only model, n is higher in metamorphic and plutonic rocks, which also tend to be more resistant (e.g., Moosdorf et al., 2018). This is not surprising given that n is thought to be influenced by higher thresholds for rock detachment, which in turn can lead to more non-linear behaviour."

Line 155: Not sure it matters for the general outcome of the study, but SRTM30 vertical errors are many times higher than for COP30. I think the whole community should move away from using SRTM data.

See comment to the point below

Line 161-164: I think it would be more beneficial if you could make this comparison with COP30 and COP90 data, or simply use those.

In response to this comment we ran the diffusion and advection-only models on the COP-30 DEM. We emphasize that the advection results are very similar and the optimized diffusion exponent is the same. However, somewhat counterintuitively, the COP-30 data gives a worse fit than SRTM for the diffusion-only model. I am fairly certain that this is largely because the diffusion model run on COP-30 is not able to capture higher erosion rates, whereas SRTM has a more heavy-tailed erosion rate distribution in models run on steep catchments. Whether this is real or simply due to noise in the SRTM, paradoxically it appears that SRTM, while having higher vertical errors, may be better able to capture roughness in steep terrains when run with the diffusion model using exponent $n=2$. I added this discussion in the Supplement. I thought some of this could be due to the processing that goes into COP-30, but I don't want to speculate too much. In particular I thought 20m was a bit of a low threshold to remove spikes, particularly for steep terranes, from what is described in Table 13 of the specifications:

 https://web.archive.org/web/20221021204908/https://spacedata.copernicus.eu/documents/20126/0/GEO1988-CopernicusDEM-SPE-002_ProductHandbook_I3.0+(1).pdf

Line 165: In general, very small catchments (< 10 km²) will be more prone to disturbances by recent mass wasting (Yanites et al., 2009). It might be worth checking how/if results change if you include those catchments.

 We have analysed this and indeed, the largest range of potential values for $n$ occurs in the smallest catchments, which as you mentioned are probably most affected by mass wasting (Figure 7). We include discussion of this on L480:

" The advection-diffusion model shows no dependence on area, but for the advection-only and diffusion-only model, the exponent values generally increase with drainage area, n ranging from ~1.1 to ~1.5, and p ranging from ~ 1.8 to 2.2. Additionally, we note the extreme range in n (~ 1.3 to > 3) for small catchments in the advection-diffusion models, potentially indicating the influence of landslides in headwaters (e.g., Yanites et al., 2010). "

Line 190: Please list the used likelihood function as an equation. As a reader, I do not want to look it up in a separate paper.

This is a good point, we have included the equation. For us it also brought to light another important point which we had overlooked. There is a case to be made regarding whether we should be referring to the coefficient of determination ($r^2$) metric here as the Nash-Sutcliffe Efficiency index instead. Although the governing equations are essentially identical, we have more often come across the use of $r^2$ when referring to regression whereas NSE is implemented in direct model comparisons, as we have done here. To minimize confusion, we have changed the reference of "$r^2$"

to "NSE" instead.  To be sure, we have used our own implementation of NSE instead of the $r^2$ as implemented in Scikit-Learn.

Are uncertainties of observed erosion rates taken into account? Do you draw normally distributed observed erosion rates, or do you use the observations uncertainty in the likelihood function? Uncertainties on the observations should be taken into account in some way.

For comparison we have implemented a version of NSE based on Harmel and Smith (2007) which takes into account measurement uncertainty (supplementary Figure S3&4).  Although this gives very similar results for optimized parameter values, it generally skews the NSE metric upwards, and is relatively new / less commonly used when compared to standard NSE.  Therefore, we have left this to the supplementary figures.

Line 230: I am not a climate specialist, but from what I get WorldClim is a bit outdated and newer, higher-resolution rasters are available (e.g. CHELSA).

Also speaking as a non-specialist in this area, it seems that although WorldClim v2 and CHELSA were released around the same time, that CHELSA is more influenced by local topography, and shows greater variability at higher resolution.  We have included CHELSA as the main dataset and left the WorldClim results out.  This slightly modifies the results of Figure 5, specifically adding more scatter at higher resolution.  Thus, we have reduced the number of bins from 40 to 20 for Figure 5e, although we keep the 40 bin results in the supplementary file (Figure S7).

Line 237: This equation includes the assumption that precipitation scales linearly with discharge. This caveat should be acknowledged.

This is a good point and we have acknowledged this on L270.

Figure 4a: I suggest to change the color map to a perceptually uniform, colorblind-friendly color map.

We agree that this is an important factor, in particular this figure panel is reliant on color for interpretation, so we have changed to a more colorblind-friendly scheme.

Figure 5: This figure is very interesting! A color bar for MAP is missing. Also, it might be more informative to have 3 panels instead of one combined. Then the uncertainties can be shown. The current representation does not allow to assess how robust this pattern with MAP really is compared to the scatter.

To minimize space used by the figure we had left the reader to rely on panel b for the color scale, but perhaps this isn't the best approach.  We have now included a colorbar.  We have added average uncertainties to Figure 5&6 and absolute uncertainties in the supplemental file (Figure S8/S9), as well as plots of residuals, emphasizing an expectedly large range in residuals but with an overall appearance of a log-normal distribution for most bins.

Section 4.3: Also these findings are very interesting. Similarly, to figure 5, having individual plots with uncertainties would give the reader a better understanding of the uncertainties in the analysis. Some of your findings here are quite surprising, such as the highly erodible carbonate rocks. Some of the general tendencies differ from similar studies relating topography to rock erodibility on a global-scale (Moosdorf et al., 2018; Ott, 2020). The findings here, should be discussed in light of previous work.

In response to this and in response to the review of Dr. Gailleton we have included more analysis of what could cause be responsible for the somewhat counterintuitive results in some of the basins. Specifically it may be that stream power is not as applicable in some of the lithologies. We have also added to the discussion with previous work. We also emphasize that there is significant variability in the model residuals within each bin, however these mostly appear log-normal (Supplemental Figure S9). We have also added a few sentences emphasizing potential confounding factors resulting in the non-intuitive results:

L471: "Other studies, for example, argue that carbonates should be less erodible (Ott et al., 2021), and that volcanic rocks may be relatively more erodible than we have shown here (Moosdorf et al., 2018). Some of these variable findings possibly arise due to factors, such as fracture density and weathering condition, which are difficult to accommodate in large-scale analyses (Neely et al., 2019), and also contribute the large uncertainty (Supplementary Figure S9)."

References: I think a mistake happened with the Marder and Gallen 2022 reference. It's cited as published in JGR:Solid Earth, when in fact it seems like it is only available as prepint on EarthArxiv. https://eartharxiv.org/repository/view/3139/

Good catch, we have corrected the reference.

References:

Lague, D., Hovius, N. and Davy, P.: Discharge, discharge variability, and the bedrock channel profile, J. Geophys. Res. Earth Surf., 110(F4), 4006, doi:10.1029/2004JF000259, 2005.

Moosdorf, N., Cohen, S. and von Hagke, C.: A global erodibility index to represent sediment production potential of different rock types, Appl. Geogr., 101, 36–44, doi:10.1016/j.apgeog.2018.10.010, 2018.

Ott, R. F.: How Lithology Impacts Global Topography, Vegetation, and Animal Biodiversity: A Global€ Scale Analysis of Mountainous Regions, Geophys. Res. Lett., 47(20), doi:10.1029/2020GL088649, 2020.

Schwanghart, W. and Scherler, D.: Bumps in river profiles: Uncertainty assessment and smoothing using quantile regression techniques, Earth Surf. Dyn., 5(4), 821–839, doi:10.5194/esurf-5-821-2017, 2017.

Whipple, K. X., Hancock, G. S. and Anderson, R. S.: River incision into bedrock: Mechanics and relative efficacy of plucking, abrasion, and cavitation, Geology, 112(3), 490–503, doi:10.1130/0016-7606(2000)112<490:RIIBMA>2.0.CO;2

Yanites, B. J., Tucker, G. E. and Anderson, R. S.: Numerical and analytical models of cosmogenic radionuclide dynamics in landslide-dominated drainage basins, J. Geophys. Res., 114(F1), 12857, doi:10.1029/2008JF001088, 2009.

Ruetenik et al. introduce and apply a method to find optimal parameter values in Landscapes Evolution Models using a global 10Be compilation. They use a Montecarlo approach to run multiple one-time-step simulations to find the values of m, n, K, D minimizing the differences between simulated erosion rates and apparent erosion rates measured by CRN. They finally group

the results by lithological and climatic domains to investigate the behaviour of optimal parameters function of these variables. I believe it offers a novel way to calibrate model parameters and even highlight interesting patterns about the behaviour of the stream power law and hillslope diffusion parameters correlated with precipitation patterns. I suggest this work is suitable for publication in esurf and will benefit the geomorphological community, however, I have two concerns about the method itself and the analysis of the results as well as other minor comments that I suggest should be addressed prior to publication. The manuscript is relatively short, which is not an issue, but could benefit from additional information and sensitivity analysis to clarify the method and its potential/limitation. Please find bellow my different comments.

Boris Gailleton

**Main concerns:**

One of the main advantages of the method according to the authors is the consideration of all the pixels in the investigated catchments, as opposed to only considering fluvial processes like Harel et al., 2016 or Marder and Gallen (2022). However, this also raises a number of questions as 10Be data is sampled at the river mouth. First, rivers are by nature only representing a minority of pixels in the catchments and are therefore under-represented statistically-speaking. Secondly, the connectivity with the sampling site will be significantly different between river processes and hillslope processes: rivers "instantly" transmit the sediments to the sampling sites while hillslopes may be disconnected. The models proposed in this study seem to assume that any change in the current topography through time is instantly transmitted to the outlet of the river, which may be true on a landscape simulated only with the stream power law and linear hillslope diffusion, but more difficultly on a real landscape. Finally, the authors present a method running LEMs on one snapshot of the (current) topography, while cosmogenic nuclides integrate larger time scales. Stochastic events (e.g., monsoon, landsliding) and/or complex sediment dynamics (e.g., recycling, residence time, intermediate traps) induce potentially significant variability in the signal (see Dingle et al., 2018 for an extreme example: https://doi.org/10.5194/esurf-6-611-2018, 2018). I acknowledge these are challenging points to include in the actual simulations - especially because the process laws of the models do not explicitly account for the sediment flux (they only express vertical topographic changes) - but I suggest it is not sufficiently discussed or addressed in the manuscript in its current form.

Thank you for this point. We indeed acknowledge that we had under-discussed the limitations of the models particularly with regard to storage of sediment, the effects of landslides, and numerous other transient effects which may influence apparent erosion rates measured from [10]Be. In response, we have significantly added to the introduction and discussion, in particular by adding a limitations section (4.5). We would like to point out that a benefit of the modelling approach, however, is its simplicity – at the global scale a more complex LEM would have to make a large number of assumptions proportional to the number of parameters. Adding parameters describing landsliding, sediment storage, etc. may not lead to more accurate results unless one were to hone in on specific regions. A main point of the discussion is copied below:

L496: "Our approach cannot account for several factors that are known to bias 10Be-derived denudation rates (e.g., von Blanckenburg, 2005; Dingle et al., 2018; Hippe et al., 2019; Struck et al., 2019). Such factors violate two key underpinning premises of the method: steady and uniform erosion across the catchment and continuous exposure of sediment at/near the surface (noted in Section 1.1). For example, sudden pulses of sediment from sources of deep-seated mass wasting can

shift apparent denudation rates downward by diluting the nuclide abundances in the river sediment sample. Given that our LEM is deterministic, it is difficult to model the impact of landslides on nuclide inventories, particularly in small headwater catchments where their influence will be greater (e.g., Yanites et al., 2010). A second pertinent issue is that our modelling assumes the erosional flux is transported instantly to the catchment outlet without intermediate storage. And yet, river sediment is likely to experience multiple episodes of erosion and deposition, especially in large lowland catchments where the volume of sediment storage expands greatly. Intermediate sediment storage can push 10Be-derived denudation rates either upwards or downwards depending on the duration and depth of sediment burial (Wittman and von Blanckenburg, 2016; Struck et al., 2018). The extent of bias in the OCTOPUS dataset (Codilean et al., 2022) is essentially unknown but it no doubt contributes to the data scatter we report here.

In addition to the complications with stream-power and diffusion-based models (noted in the Introduction) the inherent assumption that channel width increases monotonically with stream discharge (or its proxy, drainage area) is one that is violated widely. Channel width narrows at knickpoints (Whitbread et al., 2015; Yanites et al., 2018) and across diverse substrate erodibilities (Jansen et al., 2010; Croissant et al., 2017). The value of n extracted from transient landscapes will be influenced by this process, and a goal of future global analyses should be to incorporate more sophisticated rules for channel width evolution (e.g., Yanites et al., 2018)."

My other main concern is about the results, expressed in Figure 2, 3, 4, 5 and 6. It is not entirely clear whether the method picks a set of parameters, runs this same set for all the 3618 selected catchments, and calculates the global performance of this set of parameters; or if the Montecarlo process is done for each basin separately and then averaged. In other words, it is not entirely clear what these points exactly represents. In any cases, I suggest it would be relevant to show and discuss the breakdown of the global datasets: one could expect an average best parameter to be misrepresentative of the whole variability: multiple clusters could eventually emerge. This is also applicable to the Lithology and Climate comparisons: grouping metamorphic rocks into a unique category can be misleading as one would expect a very different rock strength via K (or even erosion processes via n) in a schist domain vs a gneiss domain – while both metamorphic. On the opposite, showing that the best-fit parameters are relatively constant in different contexts would be an important finding.

Thank you for bringing this up - it is true that perhaps a more ideal approach would be to find optimum parameters within each basin individually, and then describe how these parameters cluster as you mentioned. However we were unable to find a way to do this given that we only have essentially one observation per basin (erosion rate). Given the several orders of magnitude feasible range of $K$, this means that any individual rate can be fit under for a single basin given essentially any combination of $D$, $m,n,$ and $A_c$. In other words, I believe we may require more than one observation per basin to perform the optimisation in the way you suggested, which would only be (potentially) possible for the small subset of basins that have multiple nested measurements.

However, this inspired us to explore the variability of $n$ and $p$ in a more explicit way, by looking at the variability of $n$ and $p$ with respect to climate, lithology, and basin area, as several other studies before us have done. While these values generally varied close to our global averages, we found a positive relationship of $n$ and $p$ with respect to MAP. We also found an expected relationship between lithology and $n$, where higher $n$ was found in (generally) more resistant lithologies (as you mentioned, it's not a perfect way of grouping but there seems to be a pattern there). You will find now a more complete discussion of this now throughout sections 4.2-4.5.

We also include more discussion on the limitations of binning in this way, for example:

L439: "One acknowledged shortcoming of our approach (noted in Section 2.4) is that some lithologies may be over-represented in areas with higher or lower MAP. As we discuss in the next section, it is a challenge to discriminate biases due to lithology from those linked to precipitation in bins that span heterogeneous lithologies."

Other comments:

- There is a missing subsection title in the method part, when describing fluvial and hillslope processes.

Fixed

- I find the use of the term "concavity" for the non-linear hillslope diffusion equation a bit confusing as it commonly refers to river processes – as stated in the manuscript. I would recommend using "hillslope concavity" or another term to avoid any confusion.

We agree that it should be clarified – but we think perhaps concavity is more accurate a term when referring to hillslopes than rivers (in rivers, $m/n$ is not really concavity in a spatial sense but instead determined by the relationship with drainage area). We have therefore changed "concavity" in the context of SP to "$m/n$ ratio", thus allowing us to reserve the commonly used term "concavity" for hillslopes.

- The "non-linear hillslope diffusion" term can also refer to Roering et al. (1999) type of laws – that include a critical slope component in the equation. While $p$ indeed makes the equation non-linear, it would be relevant to explicitly state the difference with the more common (in my experience at least) alternatives like Roering et al., 1999.

We agree and have fixed the phrasing, referring to it directly as "exponentiated" diffusion.

- The ratio between m/n is set to 0.45. While the average is in accordance with most of the literature, this value can vary quite significantly (this is acknowledged in the manuscript, and for example Wobus et al, 2006 suggest a 0.3-0.6 range using theoretical prediction, Harel et al., 2016 0.51 +/- 0.14 based on geomorphometrics and Gailleton et al., 2021 details significant variations at global scale). I am not suggesting to rerun the full analysis for ranges of m/n, but I would recommend to at least test the sensitivity of the prediction to this ratio in at least one site or one subset of the study. I believe the study would be more robust and if the method is resilient to this ratio, it would represent a notable advantage over others.

We have acknowledged the significant variation of this parameter in spite of the global average. Unfortunately, after some tests it does not appear that the method described here is able to narrow in on the $m/n$ ratio and therefore cannot be claimed as an advantage of the method. We now discuss this in section 2.3:

L489: "An important limitation of our approach is that it fails to unravel the m/n ratio, which is known to vary widely (Gailleton et al., 2021). We ran several trials (using optimized Ac = 0.05 km2 and 20 values of linearly spaced n from 0 to 4) and found that m/n = 0.3, 0.45 and 0.6 yielded negligible difference in fit (NSE = 0.48, 0.47, and 0.45, respectively) despite the similar values of optimized n = 1.4,1.2, and 1.2, respectively (Supplementary Fig. S8). One option for improvement is to determine m/n from the river profile concavity using recent integral-based techniques (e.g.,

Harel et al., 2016, Gailleton et al., 2021) before running the suite of models. This may be especially useful for tuning the m/n ratio within specific regions."

- To echo a comment about smoothing the DEM from Reviewer #1: the location of the river network is of prime importance as it define the transition between advective and diffusive processes. However, the choice of methods and their implications is not very detailed for (i) the preprocessing of the dem to remove local minima (or smooth the data) and (ii) the flow topology (D8). These can significantly affect the location of the rivers and the proportion of river pixels, especially in the case of lower-relief areas. Another point that would deserve a mention is the flow direction. Steepest descent algorithm reduces the river as 1D series of pixels with the same area no matter the drainage area. Because the process equations used to calculate E_predicted depends on the location of the river I believe this is an important point to raise – for example mentioning alternative like D_inf or multiple flow directions which would change the ratio rivers/hillslope. Again, I am not suggesting to rerun all the analysis with a different topology, but more to address this point in the method or discussion part.

This is a good point – we have added smoothing using a simple moving window average and re-run the models, which slightly modified the results as we emphasized in the comments to Dr. Ott. We have now also include discussion on the potential for flow routing and its influence on the results, not only with reference to how they behave in reality, but also how they behave in models:

L520: "The approach described here is sufficiently robust to incorporate many different types of models. We would like to see, for instance, an exploration of the way in which drainage is routed through catchments. Different flow routing schemes can give rise to notably different drainage networks and corresponding drainage areas (e.g., Endreny et al., 2003). Here we have used unidirectional 'D8' flow, but alternatives such as multiple flow routing can produce alternative results and perspectives (e.g., Pelletier, 2010). Future efforts may test a range of flow routing methods against denudation rate in order to test their efficacy."

- While some parameters benefit from the Montecarlo "brute-force" approach, others can be estimated through other means and optimize the process – especially because this approach is applied on real topography rather than simulated ones. For example, it is possible to estimate ranges of possible m/n and Ac through geomorphometry (e.g., Mudd et al., 2018, Wobus et al., 2006). It is also possible to directly determine the location of the river network (and therefore a more natural delimitation between hillslopes and fluvial processes) and bypass the need of Ac (e.g., Clubb et al. 2014).

This is true and while we did have some of this in the discussion, we have now acknowledged this in the introduction to better set the stage. We also would like to point out that we can use these previous studies for comparison of our own optimised parameters:

L70 " While LEM parameters can be estimated via direct topographic analysis (e.g., Wobus et al., 2006, Clubb et al., 2014, Mudd et al., 2018), this approach can lead to bias, as we discuss below."

- The parameter *p* is both used for the precipitations in equation 8 and the (hillslope) concavity exponent. I suggest to at least use a capital letter P for precipitations or another notation to avoid confusion.

Excellent point which we had overlooked, we have changed it

- It is unclear in the method section whether the model considers spatial variations in K_lith, D and p (precipitation). If it does not, it can be an important point of discussion: catchments showing significant lithologic contrasts can really obscure the reading of geomorphometrics (e.g., Gailleton et al., 2021b - https://doi.org/10.1029/2020JF005970).

We do not consider lithologic variations in space, although we acknowledge this could be an important factor (in line with similar discussion from Dr. Ott).  We've included a short statement to this effect in the introduction and we've added discussion on potential bias:

L470: "Other studies, for example, argue that carbonates should be less erodible (Ott et al., 2021), and that volcanic rocks may generally be relatively more erodible than we have shown here (Moosdorf et al., 2018). Some of these variable findings possibly arise due to factors, such as fracture density and weathering condition, which are difficult to accommodate in large-scale analyses (Neely et al., 2019), and also contribute the large uncertainty (Supplementary Figure S9)."

Minor comments:

l.40: I agree. I would also add a technical limitation to the parameter values: some are interconnected - for example a value of K is only valid in a given context, and if n, m or other associated laws are modified a new K needs to be constrained.

Excellent point and we have now added this limitation

l. 43: "vital" may be a bit of an exaggeration in my opinion, maybe "natural" or "crucial"?

We have changed "vital to "crucial"

l.107: "discharge" more than "flood" variability?

We have changed it

l.109: Harel et al. 2016 also reports significant variability in this value, also suggested by Gailleton et al. 2021 from geomorphometrics.

Thank you for bringing this up, we have included references for the variability

l. 112: As long as K remains constant (can be good to remind that).

Good point and we have added this

l.139: While in line with the median in our global compilation, we also highlighted how variable this ratio could be (or at least its expression in the shape of the topography – which might not always reflect their translation into m/n when the system is significantly transient).

True – we added this caveat

l.200: It is not clear what K* represents at this point of the manuscript. Please clarify.

K* is the K before it has been corrected for log-transformation bias.  We now make more explicit note of this parameter.

l.232: missing "s" in variation.

Fixed

l.246: refer to equation.

Fixed

l.293: I disagree with this sentence: Harel et al., 2016 calculate ksn from M_chi using the method detailed in Mudd et al., 2014. M_chi is the gradient of a statistically segmented Chi-elevation River profile and is equal to ksn if A_0 = 1, otherwise proportional to it to a factor A_0. M_chi is then directly relatable to K and n within the stream power referential via the same relationship" M_chi = (E/(K * A0^m))^1/n.

We agree that the wording here was confusing and we have clarified. See the comment below which explains what we believe is a fundamental difference of the methods used.

l.295-300: while Harel et al., 2016 indeed only consider the fluvial section of the landscape, they do integrate spatial variation of ksn within a basin and calculate it for each river pixel. I suggest rephrasing/reworking the whole paragraph comparing the results with Harel et al., 2016.

We agree that the wording here was also confusing – we have clarified that one of the fundamental advantages of the approach taken here is with respect to the non-linear effects of $n =/= 1$ at the pixel scale. Based on our understanding that Harel et al. regressed the average erosion rate against one integrated ksn for each observation, which potentially biases the results although likely mostly in transient basins:

L328: "In principle, $M_\chi$ and $n$ should be similar to our $K$ and $n$ values; however, $n$ is derived from a regression of $E_{apparent}$ against $M_\chi$ and it is integrated based on each pixel within the catchment. The large discrepancy between our globally optimised values of n and those of Harel et al. (2016) may stem from the inability of the latter method to accommodate inherent nonlinearities at the sub-catchment or sub-reach scale when $n \neq 1$, whereas our approach is designed to capture some of these nonlinear effects. This is particularly important in transient catchments, as spatial heterogeneity in denudation rate is often controlled by steep areas in the catchment, such as knickpoints, and higher values of n amplify the proportion of denudation derived from steep areas relative to the rest of the catchment (Fig. 1)."

l.301: Braun and Willett, 2013 barely mention linear diffusion, I suggest to use more relevant reference(s).

This is a fair point – we now reference Forte et al., (2016) instead.

---

## Author Response (AR2)

We thank the editor and the reviewers once again for their responses.  We have addressed the comments as follows:
* * *
L25: Correlation with what? I guess you mean, minimize the misfit between measured and modolled erosion rate? Please, clarify.

We have now clarified that we are minimizing the misfit between models and observations in the abstract

L37: In your previous version, you stated „most erodbile". I would change it back to that because the actual erosion rate would mainly depend on the distribution of climate zones in respect to tectonic uplift.

Done

L376: word missing.

Fixed

Section 4.3: I would guess that the range of erodibilites is also related to the spatial scale of the analysis.

That is true, here we have clarified on line 448 that these differences may be due in part to scale.

Fig. 7: I really appreciate adding this figure. Can you please use the actual drainage areas on the x-axis tick marks. It's tough for a reader to convert the log back to the actual number (also not knowing if this is refering to the natural logarithm or log10).

Done
* * *
Maybe one last suggestion would be to remind briefly in the conclusion that the study optimises parameter values at global scales and that these can significantly vary locally. I know by experience that providing global-scale compilations of values for parameters can sometimes lead future studies to overlook the details of its variations and focus on the best-fit - however I leave that choice to the authors.

Good point, we have added this line to the conclusions:

"At the local and regional scale, optimised values will differ from what we have inferred here, but future studies may use these parameter ranges as a baseline to inform large-scale landscape evolution studies."